# Specific intracellular signature of SARS-CoV-2 infection using confocal Raman microscopy

Hamideh Salehi [1,8✉], Anuradha Ramoji [2,3,4,8], Said Mougari[5], Peggy Merida[6], Aymeric Neyret[7], Jurgen Popp[2,3,4], Branka Horvat [5,9], Delphine Muriaux[6,7,9] & Frederic Cuisinier [1,9]

SARS-CoV-2 infection remains spread worldwide and requires a better understanding of virus-host interactions. Here, we analyzed biochemical modifications due to SARS-CoV-2 infection in cells by confocal Raman microscopy. Obtained results were compared with the infection with another RNA virus, the measles virus. Our results have demonstrated a virus-specific Raman molecular signature, reflecting intracellular modification during each infection. Advanced data analysis has been used to distinguish non-infected versus infected cells for two RNA viruses. Further, classification between non-infected and SARS-CoV-2 and measles virus-infected cells yielded an accuracy of 98.9 and 97.2 respectively, with a significant increase of the essential amino-acid tryptophan in SARS-CoV-2-infected cells. These results present proof of concept for the application of Raman spectroscopy to study virus-host interaction and to identify factors that contribute to the efficient SARS-CoV-2 infection and may thus provide novel insights on viral pathogenesis, targets of therapeutic intervention and development of new COVID-19 biomarkers.

[1] LBN, University of Montpellier, Montpellier, France. [2] Friedrich-Schiller-University Jena, Institute of Physical Chemistry and Abbe Center of Photonics (IPC), Helmholtzweg 4, D-07743 Jena, Germany. [3] Leibniz Institute of Photonic Technology (IPHT), Member of Leibniz Health Technologies, Albert-Einstein-Straße 9, D-07745 Jena, Germany. [4] Jena University Hospital, Center for Sepsis Control and Care (CSCC), Friedrich-Schiller-University Jena, Am Klinikum 1, 07747 Jena, Germany. [5] CIRI, International Center for Infectiology Research, INSERM U1111, CNRS UMR5308, Université de Lyon, Université Claude Bernard Lyon, École Normale Supérieure de Lyon, Lyon, France. [6] Institute of Research in Infectiology of Montpellier (IRIM), University of Montpellier, UMR9004 CNRS Montpellier, France. [7] CEMIPAI, University of Montpellier, UMS3725 CNRS Montpellier, France. [8] These authors contributed equally: Hamideh Salehi, Anuradha Ramoji. [9] These authors jointly supervised this work: Branka Horvat, Delphine Muriaux, Frederic Cuisinier. ✉email: s_hamideh@yahoo.com

Coronavirus disease 2019 (COVID-19) is an ongoing pandemic infection caused by the positive-sense RNA virus, severe acute respiratory syndrome coronavirus 2 (SARS-CoV-2), provoking untold disruption throughout the world. Symptoms of COVID-19 such as dyspnea, fever, cough, and fatigue could be followed by numerous complications including pneumonia, myocarditis, and kidney injury, resulting in some cases of death[1]. Understanding the mechanisms of virus-induced cell modifications is critical for the development of rapid diagnostics of infection and efficient antiviral treatment.

Sensitive viral identification could be obtained using currently available methods, including immunological and molecular tests, ELISA (enzyme-linked immunosorbent assay), and PCR (polymerase chain reaction)[2–5], however, these analyses require previous information on the infectious agent identity[6]. Other techniques such as next-generation sequencing are very sensitive, although low virus quantity may be challenging[7,8]. Globally, 60% of medical care is engaged by viral infections[9], and the requirement of accurate biomarkers of viral infection presents a big challenge. Due to the significant heterogeneity of COVID-19 disease profiles, biomarkers that allow either the identification of patients at high risk for developing severe forms of COVID-19 and its long-term complications, or guide personalized treatment options, are scarce. Therefore, new approaches leading to the identification of COVID-19-related biomarkers are urgently required for the development of precision medicine-based therapeutic strategies.

Cells could respond to different steps of viral infection, including virus entry, trafficking through the cell organelles for replication and egress of viral particles, or their entry into lysosomes for degradation of viral proteins[10,11]. As Raman microscopy allows the analysis on the single-cell level[12–14], the biochemical modifications due to the cell reaction to the virus could be detected in the early stage. The conventional diagnostic methods are applicable after symptoms appear which might be too late to avoid serious consequences. As Raman identifies the chemical modification, by investigation of spectral changes induced by a virus in any environment, there is no requirement for any genetic or proteomic information of the virus[7,15–17], presenting an additional advantage for the Raman spectroscopy in viral detection.

Biomedical applications of Raman spectroscopy have been reported previously due to its noninvasive examination of a single live cell without labeling[14,18–20]. Raman spectroscopy imaging is known as a well-established method for chemical analysis of intra/extracellular environment[21,22]. Each confocal Raman voxel contains the biochemical fingerprint signature with a dimension of 300 nm × 300 nm × 900nm[19,20]. Data analysis on this information reveals precise cell comportments identifications as well as slight modifications. Usual data analysis on Raman spectra is categorized into two groups supervised and unsupervised approaches[14]. The unsupervised methods only depend on Raman spectra to classify into different categories. By reducing the number of variables, principal component analysis (PCA) is considered as an unsupervised method. As a supervised method to differentiate between various cell organelles (including the nucleus, nucleolus, mitochondria, Golgi, and cytoplasm), k-means cluster analysis (KMCA) could be realized on more than twenty thousand spectra of each Raman scan[19,23]. Organelle's separation enables distinctive extraction of single spectra for further hyperspectral data analysis such as PCA or support vector machine (SVM). Viral infection with different incubations time has been monitored in previous studies[24,25]. Different viral dilutions to mimic saliva conditions were been used to detect RNA virus using Raman spectroscopy[26]. Recent studies reported the detection of SARS-CoV-2 using surface-enhanced Raman

scattering (SERS)[1,27–32]. SERS results coupled with PCA analysis is been introduced as a powerful tool to detect the virus[33]. In this study gold nanoparticles are been used as a substrate to create the plasmons for surface-enhanced Raman signals on transfected HEK293 cells. SERS substrate design dictates sensitivity and accuracy of the method and presents an essential key towards a successful result, but is also associated with a rather high cost of analysis[1,29–31].

In the current study, spontaneous Raman spectroscopy has been used to avoid complex substrate design. Biomolecular modification of cellular organelles after SARS-CoV-2 infections has been compared to another RNA virus, the measles virus (MeV) to analyze virus-specific intracellular modifications. The list of obtained biochemical changes is provided to help in better understanding the cellular signature of virus infection. The obtained results show tryptophan traces in SARS-CoV-2-infected cells which were observed neither in noninfected cells nor in MeV-infected cells. These results suggest an easily detectable viral signature in an early stage of infection and a potentially new biomarker of COVID-19. Understanding the intracellular biochemical modification following the virus infection of the host cells should provide new information about SARS-CoV-2-induced intracellular modifications and may reveal specific markers for personalized treatment of COVID-19 patients.

## Results

**Raman imaging of virus-infected Vero E6 cells.** SARS-CoV-2-induced modifications in different sub-cellular compartments were followed in Vero E6 cells, a widely used cell line for the amplification of different viruses. Cells were initially infected with SARS-CoV-2 or MeV for 24 h and analyzed by Raman spectrometry. This time point was determined in the initial analysis as optimal to observe the effect of viral infection on cells. The SARS-CoV-2 infection of Vero E6 cells was verified by immunoblots, immunofluorescence, and electron microscopy (Supplementary Fig. S1). Immunoblots anti-ACE2 show the presence of the virus receptor expressed in Vero E6 cells, compatible with SARS-CoV-2 infection, before and after viral infection, in contrary to HEK293T cells that do not harbor ACE2 receptor. Anti-Spike and anti-N antibodies reveal the abundant expression of the CoV-2 Spike and Nucleocapsid N proteins in cell lysates of infected cells at MOI = 0.1 (Supplementary Fig. S1A). In parallel, the sample of SARS-CoV-2-infected Vero E6 cells were analyzed by immunofluorescence at 24 and 48 h post-infection showing that the majority, if not all, of the cells were infected by SARS-CoV-2 (Supplementary Fig. S1B). By electron microscopy, we also follow the virus infection and observe the visible intracellular modifications due to the virus, 24 h post-infection (pi) (Supplementary Fig. S1C). Transmission electron microscopy (TEM) images showed viral particles at the cell plasma membrane signing viral production (Supplementary Fig. S1C-a) and/or further virus attachment for cell entry. In addition, intracellular organelles filled with viral factories were recognizable inside the cytoplasm, near the nucleus and mitochondria, most probably reflecting the double-membrane vesicles or spherules replicative factories of SARS-CoV-2 derived from the ER-Golgi apparatus (Supplementary Fig. S1C-b) as previously observed and described[34,35]. These TEM images confirmed that virus particles were produced by the infected cells and that they can be visualized at the cell plasma membrane and 24 h post-infection.

We then proceeded with Raman microscopy to follow virus-induced intracellular biochemical modifications traceable in sub-cellular organelles using a label-free method. For Raman imaging of virus-infected cells and measurements, the central wavelength of gating was set at 610 nm (which presents the common setting for Raman cell imaging), showing the "fingerprint region" from 400 to 1800 cm$^{-1}$, the C-H peak at 2800–3100 cm$^{-1}$. Each

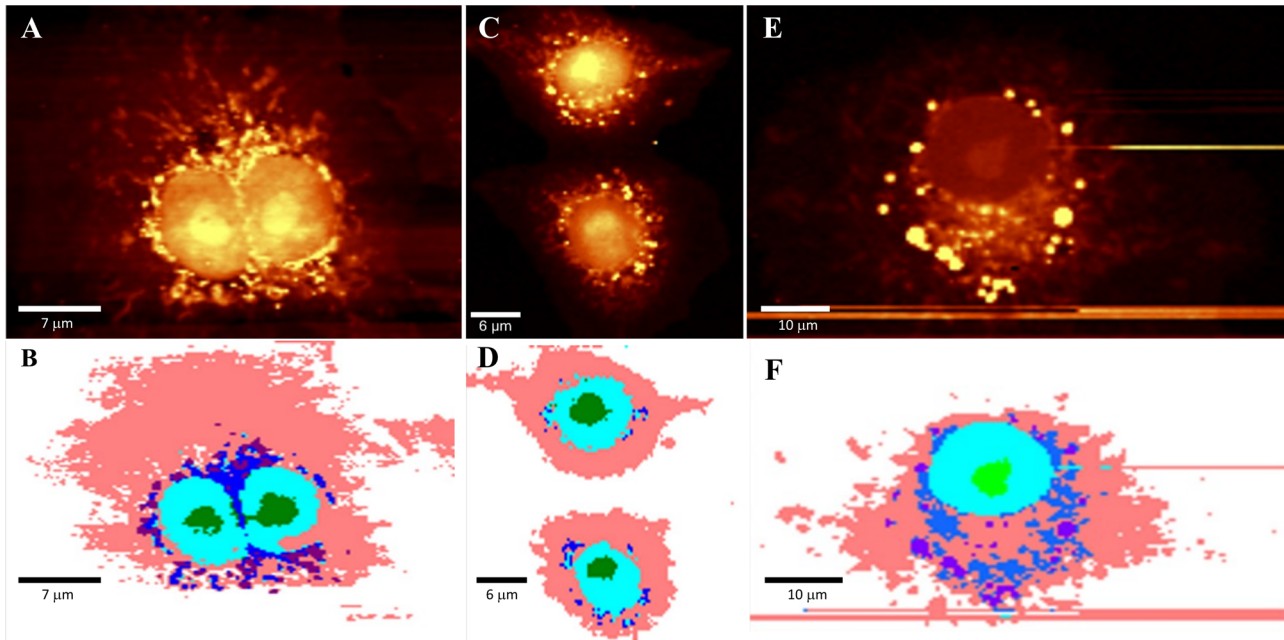

**Fig. 1 Raman confocal images of SARS-CoV-2 and MeV-infected (24 h pi) and noninfected Vero E6 cells. A** Raman images of noninfected cells Light yellow color corresponds to the highest intensities of lipids and proteins and dark shadow for the lowest intensities. **B** KMCA results for the noninfected cells of part A. The nucleolus is marked in dark green, the nucleus in turquoise, mitochondria and Golgi are in blue, cytoplasm in pink and lipid droplet in purple. The same color code has been used for all analyses. **C** SARS-CoV-2-infected Raman C-H images and **D** relevant KMCA illustrations for the C part. **E** MeV-infected Raman C-H images and **F** relevant KMCA illustrations for E part. Scale bars are variable between 6–10 μm as indicated.

Raman scan contains more than twenty thousand individual spectra. A Raman reconstructed image is based on the selected C-H bands to represent the intracellular proteins and lipids. During the scan, with the nanometric scale movement of the piezoelectric table, each pixel is registered with its relevant spectrum. Molecular structure information is gathered in Raman spectra with individual bands associated with the biomolecules under the laser spot. The average spectral shape of K-means cluster analysis (KMCA) clusters was obtained to represent the cellular organelles, such as the nucleus, cytoplasm, mitochondria, and Golgi. To assure the virus specificity of observed effects the results obtained with SARS-CoV-2 were compared with those found after the infection of Vero E6 cells with another RNA virus, the measles virus (MeV).

Figure 1 illustrates scans of SARS-CoV-2- and MeV- infected for 24 h and noninfected Vero E6 cells followed with the relevant KMCA analysis. Cellular organelles separation is followed after KMCA analysis to extract the single spectrum of each cluster. Figure 1A, C, E illustrates reconstructed C-H Raman images of infected versus noninfected cells, with light yellow as the highest intensities and dark hues for the lowest intensities of the C-H peak, corresponding to the intracellular content of lipids and proteins. Part 1B-D-F belongs to the relevant KMCA analysis of 1A-C-E. This analysis allows to mark the cellular organelles to separate single spectra of each organelle (nucleus, nucleolus, Golgi-mitochondria together, lipid droplets) as well as the cytoplasm, for further principal component analysis (PCA) and support vector machine (SVM) analysis. For a better comparison, the same color code has been used for all KMCA parts as indicated in the legend.

**Chemical profile of SARS-CoV-2 and MeV-infected Vero E6 cells**. The mean of the Raman spectra extracted from different intracellular components, namely, cytoplasm, Golgi-mitochondria bodies, and nucleus region of noninfected and SARS-CoV-2/MeV-infected Vero E6 cells has been displayed in

Fig. 2A–C, respectively. For visualizing the difference between intracellular components as a result of the viral infection, the Raman difference spectra were generated by subtracting the mean Raman spectra of the noninfected cells from the mean Raman spectra of the cells infected with the virus (the Raman spectra from the cells with MeV infection and cells with SARS-CoV-2 infection were used together in the infected group), for individual intracellular components, as shown in Fig. 2D–F. Further, for a deeper insight into the biochemical difference between MeV and SARS-CoV-2-infected cells, the Raman difference spectra were generated for different intracellular components by subtracting Raman spectra of the SARS-CoV-2-infected cells from the MeV-infected cells (Fig. 2G, H).

In the difference spectra shown in Fig. 2D–F between noninfected (Control) and the virus infection (MeV and SARS-CoV-2 infection considered together), the positive Raman peaks belong to the control group, whereas the negative Raman peaks belong to the virus-infected cells. The Raman difference spectra (Fig. 2D–I) highlights the intracellular changes occurring within the cell as a result of viral infection. The differences between infected and noninfected cells in the cytoplasm, Golgi-mitochondria bodies, and nucleus can be observed mainly in the Raman peak profile of the C-H stretching region (2800–3050 cm$^{-1}$) (Fig. 2D–F). In the difference spectra of cytoplasm and Golgi-mitochondria bodies shown in Fig. 2D, E, phosphate backbone vibration is visible at 788 and 790 cm$^{-1}$ respectively[36], possibly due to the presence of viral RNA[37]. The Raman spectra extracted from the noninfected cells' Golgi-mitochondria bodies show a Raman spectral signature of lipid at 1732 cm$^{-1}$ [38] (Fig. 2E). The difference in the Raman spectrum between the two types of viral infections, MeV and SARS-CoV-2, indicates chemical changes that are specific to the type of the virus causing the infection (Fig. 2G–I). The positive Raman peaks in the presented spectra belong to the SARS-CoV-2-infected cells and the negative Raman peaks belong to the MeV-infected cells. Within all the three intracellular compartments, SARS-CoV-2-infected cells show the presence of symmetric CH$_2$ vibrations of

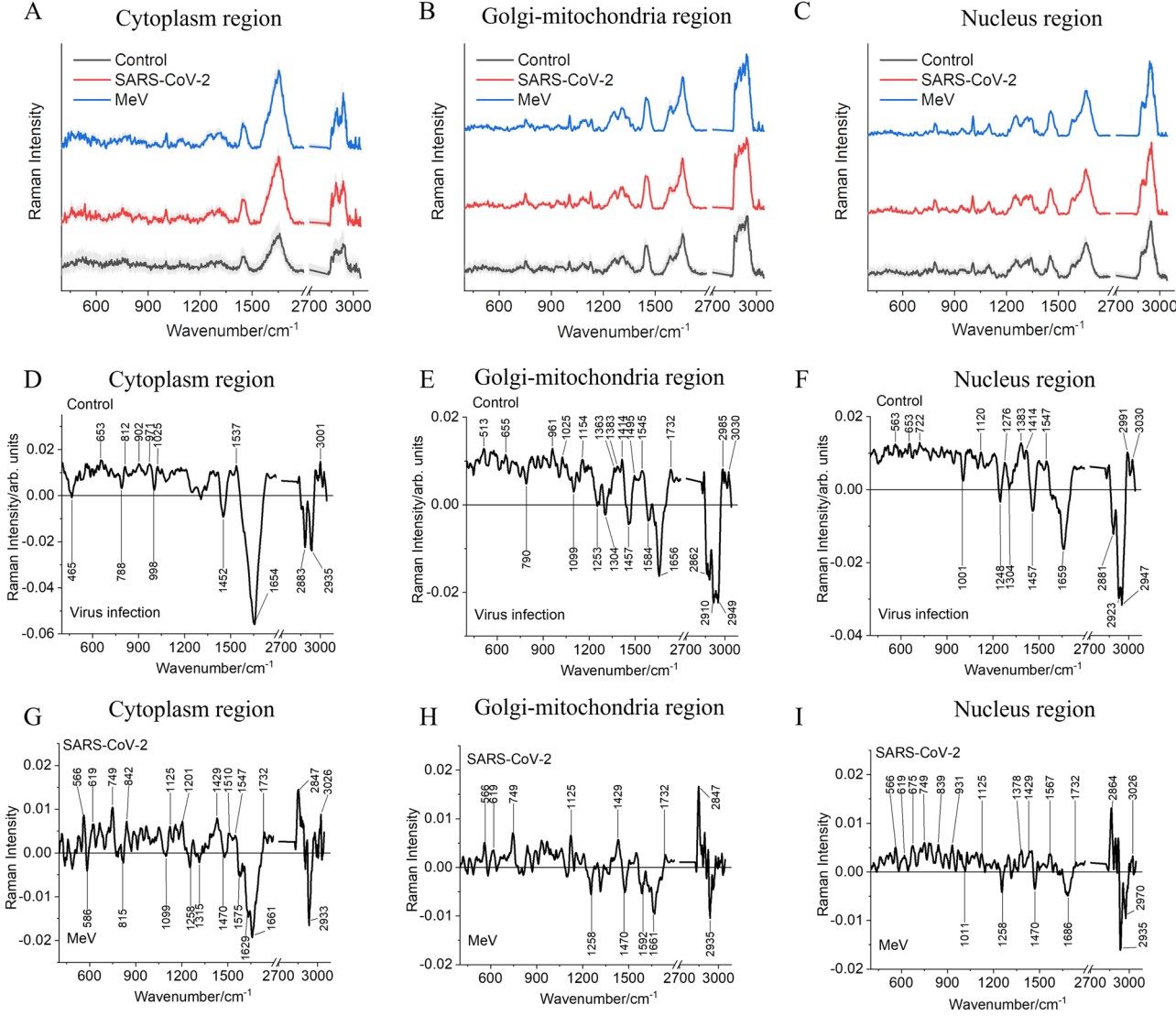

**Fig. 2 Mean Raman spectra from the cytoplasm, Golgi-mitochondria bodies, and nucleus region shown for noninfected (Control) and infected Vero E6 cells (SARS-CoV-2 and MeV) along with the standard deviation.** **A** Raman spectra were extracted from the cytoplasm region, **B** Raman spectra were extracted from Golgi-mitochondria bodies, and **C** Raman spectra were extracted from the nucleus region. **D–F** Difference Raman spectra calculated between noninfected (Control) and virus-infected cells (MeV and SARS-CoV-2 taken together) for **D** cytoplasm, **E** Golgi-mitochondria, and **F** nucleus region of the Vero E6 cells. **G–I** Difference Raman spectra calculated between the SARS-CoV-2-infected cells and MeV-infected cells and for **G** cytoplasm, **H** Golgi-mitochondria, and **I** nucleus region of the Vero E6 cells. The difference Raman spectra have been smoothed using the Savitzky–Golay method with points of window 15 and polynomial order 2. The Raman spectra are shifted on the y-axis for sake of clarity.

lipids, (1732 cm⁻¹ and 2847/2864 cm⁻¹)[39,40], ring breathing modes of the DNA/RNA bases (1429 cm⁻¹)[41], strong C-O band of ribose serves as a marker of RNA (1125 cm⁻¹)[38], tryptophan vibrations (566 and 749 cm⁻¹)[42–45], C-C twisting mode of phenylalanine (619 cm⁻¹)[44], whereas MeV-infected cells show dominating presence of proteins (2933/2970 cm⁻¹)[40]. The assignments of the Raman peaks have been described in Table 1 and Supplementary Table S1.

To visualize the possibility to differentiate between the noninfected, MeV-infected, and the SARS-CoV-2-infected Vero E6 cells, principal component analysis (PCA) was performed. The PCA 3D-score plot along with the principal component (PC) loadings are shown in Fig. 3 for the three different intracellular components. The PCA analysis was performed for each experimental batch separately to evaluate the Raman spectral data. In Supplementary Fig. S2, PCA analysis for the remaining

two batches has been shown. In the PCA 3D-score plots shown for cytoplasm (Fig. 3A), Golgi-mitochondria bodies (Fig. 3C), and nucleus (Fig. 3E), the Raman spectra of noninfected Vero E6 cells are well separated from the Raman spectra of the infected cells. This indicates a distinct chemical profile between infected and noninfected cells as shown in Table 1 and Supplementary Table S1. The Raman spectra extracted from the nucleus region of the MeV and SARS-CoV-2-infected cells (Fig. 3E) are well separated from each other, compared to the Raman spectra extracted from the cytoplasm and Golgi-mitochondria bodies of infected Vero E6 cells. This indicates the nucleus of MeV-infected Vero E6 cells has a different chemical profile compared to the nucleus of the SARS-CoV-2-infected Vero E6 cells. The PC loadings are shown in Fig. 3B, D, F show the respective contributing Raman peaks. The PC1 is mainly responsible for separating infected and noninfected Vero E6 cells. The PC2 and

**Table 1 Significant Raman peaks observed in PC loadings contributing to the separation between SARS-CoV-2 and MeV-infected Vero E6 cells[24-26,33,53,60].**

Raman peaks in cm$^{-1}$

| Cytoplasm | | Golgi-mitochondria- | | Nucleus | |
|---|---|---|---|---|---|
| **MeV** | **SARS-CoV-2** | **MeV** | **SARS-CoV-2** | **MeV** | **SARS-CoV-2** |
| 586 (sym str PO$^3_4$) | 459 (DNA/Glycogen) | 1258 (Ade/Cyt) | 566 (Trp/Cyt/Gua) | 733 (Ade) | 566 (Cyt/Gua) |
| 815 (Tyr) | 566 (Trp/Cyt/Gua/Kyn) | 1470 (lipids) | 619 (Cys) | 1011 (Str C-O Ribose) | 675 (Gua) |
| 1099 (str C-N) | 749 (Trp) | 1592 (Str C = C and C = N protein) | 749 (Trp) | 1258 (Ade/Cyt) | 749 (Tyr) |
| 1250 (Gua/Cyt) | 842 (Trp/Kyn) | 1661 (α-helix protein) | 1125 (Trp/Str C-C lipids) | 1279 (Nucleic acids/ Phosphates) | 839 (Pro/hydroxyproline, Tyr/glycogen) |
| 1315 (Gua) | 1125 (Str C-C lipids/glucose/ polysaccharide) | 2935 (CH$_3$ sym str proteins) | 1429 (Deoxyribose) | 1289 (Cyt) | 931 (Gua) |
| 1487 (Gua) | 1201 (Trp) | | 1732 (lipids) | 1345 (Gua) | 1125* C-O ribose |
| 1575 (Gua/Ade) | 1429 (Fatty acids) | | 2847 (CH$_3$ sym str lipids) | 1470* (lipids) | 1235*PO$_2$ mode |
| 1629 (Tyr) | 1510 (Cyt) | | | 1641* (proteins) | 1378 (Ade/Gua) |
| 1661 (α-helix protein) | 1547 (Trp) | | | 1686* (proteins) | 1429 (Deoxyribose) |
| 2939 (lipids) | 1607 (Trp) | | | 2935* (proteins) | 1477 (Gua) |
| | 1732 (lipids) | | | 2970* (lipids) | 1575 (Gua/Ade) |
| | 2849 (CH$_3$ sym str lipids) | | | | 1732* (lipids) |
| | 3026 (long-chain fatty acid) | | | | 2864* (lipids) |
| | | | | | 3026 (long-chain fatty acid) |

*Trp* tryptophan, *Cys* cysteine, *Tyr* tyrosine, *Cyt* cytosine, *Gua* guanine, *Pro* proline, *Sym* symmetric, *Str* stretching, *Ade* adenine, *Kyn* kynurenine.
*The contribution of these Raman bands arises due to the extraction of nucleic acid spectra from the Raman spectroscopic image of the cells.

PC3 jointly contribute to the separation of the MeV- and SARS-CoV-2-infected Vero E6 cells. Further, in the cytoplasm of MeV, infected cells presence of tyrosine (815 and 1629 cm$^{-1}$) is specifically observed compared to the SARS-CoV-2-infected cells where tryptophan peaks (755, 1201, 1547, 1607 cm$^{-1}$) are prominent (Table 1).

**Raman model to differentiate SARS-CoV-2 and MeV-infected Vero E6 cells**. The unsupervised PCA was unable to resolve the infected and noninfected Vero E6 cells when Raman spectral data from all three experimental batches were pooled together for the analysis. Hence, a supervised machine learning algorithm-SVM was used for the classification of the noninfected, MeV-infected, and SARS-CoV-2-infected Vero E6 cells. Two-class SVM models were built as shown in Fig. 4A–C to visualize the chemical differences. Further, Raman difference spectra were generated as shown in Fig. 4D–F. In Fig. 4A, the SVM model separates between noninfected and MeV-infected Vero E6 cells with a 10-fold cross-validated accuracy of 99%. Within the SVM plot, the separation of Raman spectra extracted from the different intracellular components of infected and noninfected cells can be visualized from the shape of the legend used (Square = cytoplasm, circle = Golgi-mitochondria bodies, and triangle = nucleus region). The difference spectrum (Fig. 4D) generated by subtracting the mean Raman spectrum of noninfected cells (positive Raman peaks) from the mean Raman spectrum of MeV-infected cells (negative Raman peaks) indicates major chemical differences observed in the infected cells are lipids (2860 and 2885 cm$^{-1}$) and proteins (1599, 2929, and 2947 cm$^{-1}$), nucleic acids (793 and 1255 cm$^{-1}$), and carbohydrates (1464 cm$^{-1}$). Similarly, the SVM model classifies the noninfected and SARS-CoV-2-infected cells with a tenfold cross-validated accuracy of 97% (Fig. 4B). The changes observed in the Raman difference spectrum (Fig. 4E) are similar to those described above for the MeV-infected cells. In Fig. 4C, the two-class SVM classification model shows differentiation between the MeV-infected cells and SARS-CoV-2-infected cells. All the three intracellular components of the MeV and SARS-CoV-2-infected cells were well separated. The total tenfold cross-validated accuracy was 98%. The difference spectrum generated between MeV and SARS-CoV-2-infected cells shows differences in proteins (1612 cm$^{-1}$ in SARS-CoV-2 and 1656 cm$^{-1}$ in MeV-infected cells), higher lipids in SARS-CoV-2 (2856 and 2885 cm$^{-1}$), changes in the nucleic acids (1091 and 1255 cm$^{-1}$ in MeV) and tryptophan vibrations at 749 cm$^{-1}$ in SARS-CoV-2-infected cells).

A three-class SVM model was built (Supplementary Fig. S3) to classify noninfected and MeV-infected and SARS-CoV-2-infected cells. The noninfected cells are well separated, further, the separation between the two types of viral-infected cells can also be visualized.

## Discussion

Understanding the virus-induced biochemical modifications in host cells is of high interest to a better comprehension of virus-host interactions and developing diagnostic biomarkers and therapeutic approaches. As Raman spectroscopy can detect these intracellular molecular metabolic changes without any marker or sample preparation[46,47] or need for previous knowledge of virus strains, this method has the potential to be used as a precise label-free detection technique with a low cost of analysis[48,49].

In this study, two highly contagious airborne RNA viruses were used to infect Vero E6 cells and compared to noninfected cells using the Raman spectroscopy approach. The initial observation of SARS-CoV-2 infection in Vero E6 cells was confirmed by immunoblots and by immunofluorescence confocal microscopy that indeed was showing that at 24 h pi almost all the cells were infected by SARS-CoV-2. By TEM microscopy, we confirmed the abundant virus replication at 24 h post-infection for an MOI equal to 0.1, revealing intracellular viral factories, and viral particle egress at the cell membrane (Supplementary Fig. S1). Raman

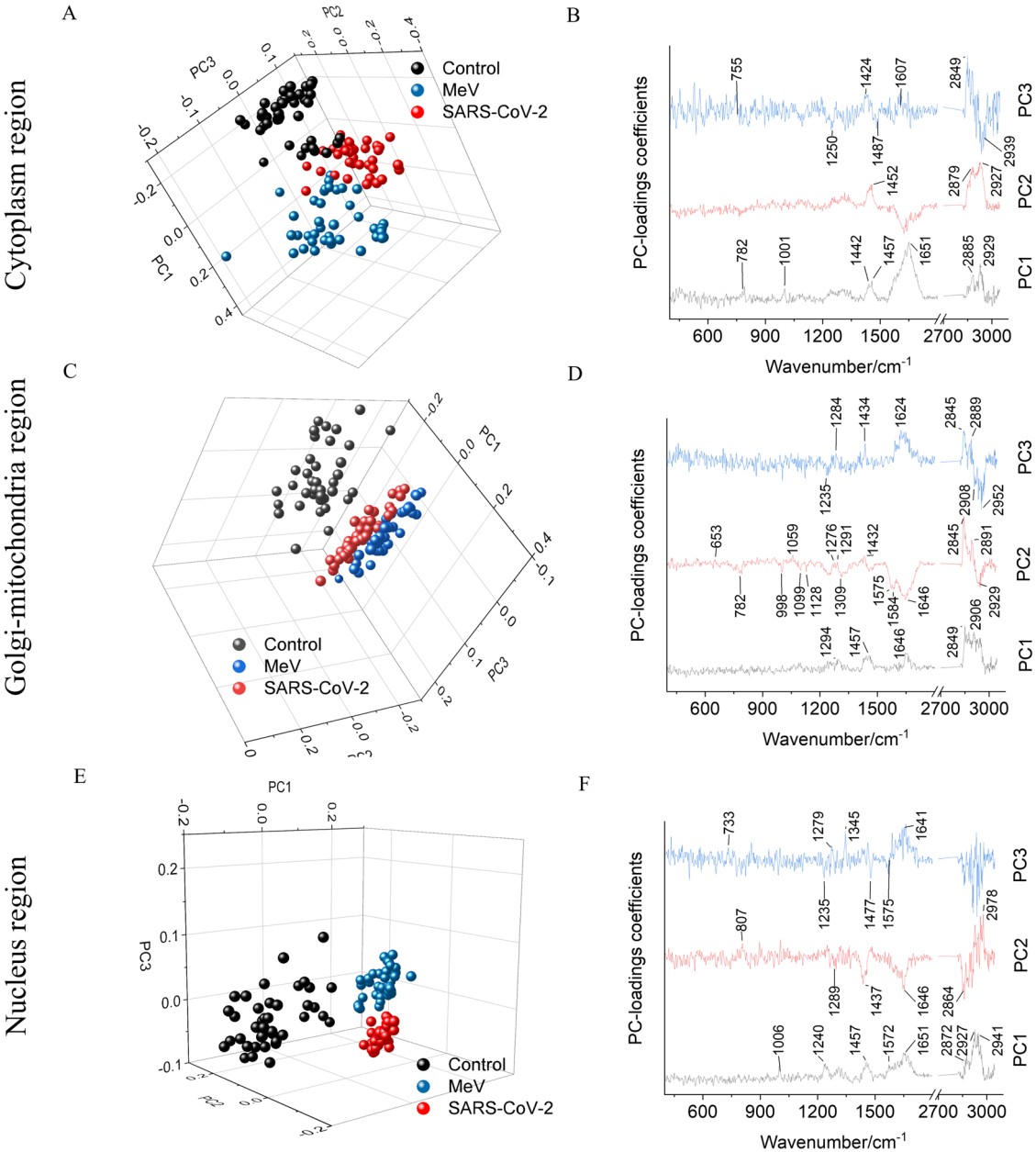

**Fig. 3 Principal component analysis (PCA) of Raman spectra extracted from intracellular components of SARS-CoV-2 and MeV-infected and noninfected (Control) Vero E6 cells.** 3D PCA score plot of Raman spectra from **A** cells' cytoplasm and **B** corresponding PC loadings, **C** cells' Golgi-mitochondria bodies, **D** corresponding PC loadings, **E** cells' nucleus, and **F** corresponding PC loadings. The loading coefficients are shifted on the y-axis for sake of clarity.

spectroscopy imaging using 532 nm laser excitation, enables along with visualization of the virions also the intracellular molecular modifications due to the effect of SARS-CoV-2 or MeV infection of the cells. Multivariate analysis is indispensable to realize major dissimilarities within the individual spectrum, hence, to understand the chemical changes occurring in the infected cells chemometric methods, namely, KMCA, PCA, and SVM were employed. The KMCA of Raman spectral data allows visualization of intracellular organelles in the micrometer ranges enabling sensing of clustered virion particles or replicative viral factories in vesicles. For proof of localization of viruses or viral replicative organelles within the infected cells, the immunofluorescence staining method was used (Supplementary Fig. S1B), since cells were intact after Raman spectroscopy imaging as it is a non-destructive method[22].

The study aimed at the identification of virus-specific intracellular biochemical changes occurring during the infection with two different RNA viruses. The Raman difference spectrum generated between noninfected/infected cells and between SARS-CoV-2/MeV-infected cells for different organelles (Fig. 2) demonstrates the unique host cell response to of SARS-CoV-2. The contributing Raman peaks as observed in Fig. 2D–I and the biomolecules giving rise to these peaks (Table 1 and Supplementary Table S1) provide hints about biochemical changes occurring during viral infection. Especially, the prominent presence of tryptophan as a significant consequence during SARS-CoV-2 infection is observed. This observation corroborates our analysis of virus genomes, revealing that the SARS-CoV-2 proteome contains approximately three times higher tryptophan levels compared to the MeV proteome (159 versus 57 aa,

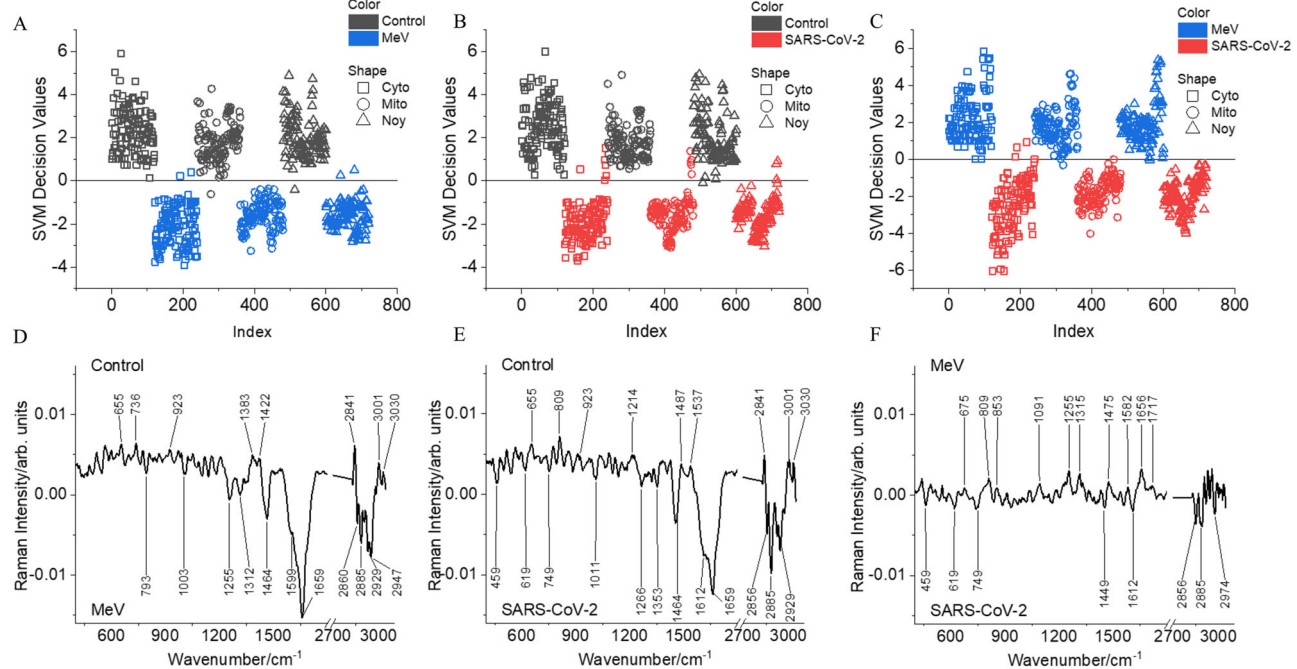

**Fig. 4 Raman model generated using support vector machine (SVM) algorithm to differentiate. A** noninfected (Control, black) from measles virus (MeV, blue) infected Vero E6 cells, **B** noninfected (Control, black) from SARS-CoV-2 (red) infected Vero E6 cells, and **C** MeV and SARS-CoV-2-infected Vero E6 cells. The model was generated using Raman spectra extracted from intracellular components: cytoplasm (Cyto, square), Golgi-mitochondria bodies (Mito, circle), and nucleus (Noy, triangle) of the Vero E6 cells (Total accuracy **A** 98.89%, **B** 97.22%, **C** 97.78%, PCs used 13, tenfold CV other parameters same as below). The difference Raman spectra calculated between **D** control minus measles virus (MeV), **E** control minus SARS-CoV-2, and **F** Measles (MeV) minus SARS-CoV-2.

respectively, found in each virus), which probably leads to a higher content of intracellular tryptophan during SARS-CoV-2 replication. Tryptophan metabolism presents an important biochemical pathway, implicated in inflammation, generation of immune tolerance, and mental health[50]. Interestingly, in accord with our results on the intracellular tryptophan changes based on Raman spectroscopy, the modifications in the tryptophan ratio in the sera of COVID-19 patients were observed in the previous studies[51,52] and the assessment of tryptophan was recently suggested to be the prognostic marker in COVID-19 patients[51]. Our results suggest that accumulation of this essential amino acid in SARS-CoV-2-infected cells may lead to a decrease in the serum level of tryptophan in infected patients, associated with aggravation of their clinical symptoms, underlying the potential of tryptophan to serve as a biomarker for COVID-19. Similar observations have been made in a recent Raman spectroscopy study on Epstein-Barr virus infection of glial cells where tryptophan along with other biomolecules such as cholesterol, glucose, and polysaccharides are associated with the viral infection[49].

Further, SARS-CoV-2 triggers different chemical reactions within the cells compared to another RNA virus MeV, which could be evidenced by the simple unsupervised PCA analysis (Fig. 3). The PCA analysis indicates that SARS-CoV-2-infected cells have different biochemical makeup compared to MeV-infected cells, demonstrating the specificity of the Raman approach. In this study only the Vero E6 cell line was used for the viral infection, to obtain the first insight into the possible application of the Raman approach that has been demonstrated to be powerful in virus discrimination. However, the use of the other cell types known as hosts for the COVID-19 infection in patients should be performed in the future and may reveal the additional fingerprints of the SARS-CoV-2 biochemical modifications in specific cell types.

For robust analysis, the experiments have repeated a minimum of three times, and the Raman spectral data were pooled together.

A higher-end chemometric method was required as PCA was unable to handle the heterogeneity within the data set normally present due to the biological and experimental repetitions. However, to avoid overfitting, the variance captured by the first 13 PCs was used as input for the SVM analysis. The differentiation of noninfected and infected Vero E6 cells visualized in the SVM decision value plot indicates the possibility to apply Raman firstly to detect the presence of infection. Secondly, the differentiation of the SARS-CoV-2 and the MeV-infected cells highlights the possibility to classify the type of virus causing the infection. The simple difference spectrum generated between SARS-CoV-2 and MeV-infected cells (Fig. 4F) for spectral interpretation of the SVM differentiation (Fig. 4C) indicates changes in the protein profile based on the Raman signature of different proteins observed in SARS-CoV-2 and MeV-infected cells. This difference in the protein profile of SARS-CoV-2-infected and MeV-infected cells indicates the differential transcriptomic signature of these two viral infections.

Huge pressure on medical establishments due to the epidemic of COVID-19 leads to the development of new detection methods which require fewer material resources and background knowledge, specifically in the case of new viral infections and their different variants, and requires the identification of new biomarkers which could allow better stratification of COVID-19 patients and allow their personalized treatment. Virus-cell interaction brings new insight into this field, based on Raman spectroscopy approaches. Developing a new data-reading method called Raman barcode, shows the precision of Raman signature to detect changes in isolated virus strain[53]. The study of spike RBD receptor interaction based on protein antibody recognition has been possible by lowering the detection limit using SERS[54]. Development of a label-free, low-cost detection method based on the significant Raman peak, indicated in this study, the testing method could be available in medical centers. The results are

obtained immediately and no previous knowledge of the virus strain is needed. The accuracy, sensitivity, and specificity are high enough in the early interaction of the virus with host cells and could be used in the case of suspected viral infection and search for the identification of biomarkers of viral infection. Finally, Raman spectroscopy allows the identification of biochemical markers of viral infection, presenting thus a promising technique in the analysis of viral infections. The presented study demonstrates the proof of concept for the application of confocal Raman microscopy combined with advanced data analysis, to study SARS-CoV-2 infection in cell culture and possibly develop COVID-19-related biomarkers. The spectral changes were followed after KMCA analysis on different cellular organelles, and separation of relevant single spectra for hyperspectral analysis, opening innovative perspectives for further research in the field.

## Methods

**Virus, infection, and cell culture**. The Vero E6 cell line (African green monkey kidney cells) were obtained from SIGMA #85020206 and maintained in Dulbecco's minimal essential medium (DMEM) supplemented with 10% heat-inactivated fetal bovine serum (FBS) at 37 °C with 5% $CO_2$. SARS-CoV-2 isolate BetaCoV/France/IDF0372/2020, was supplied by the National Reference Center for Respiratory Viruses hosted by Institute Pasteur (Paris, France). The human sample from which this strain was isolated has been provided from the Bichat Hospital, Paris, France. Moreover, the BetaCoV/France/IDF0372/2020 strain was supplied through the European Virus Archive goes Global (EVAg) platform, a project that has received funding from the European Union's Horizon 2020 research and innovation program under grant agreement No 653316. The virus was propagated in Vero E6 cells in a DMEM medium containing 2.5% FBS at 37 °C with 5% $CO_2$ and harvested 72 h post-inoculation, and titered as described previously[55]. Vero E6 cells were plated on Raman glass slides for 24 h and then infected with $10^6$ PFU of SARS-CoV-2 for 24 h multiplicity of infection (MOI) of 0.1. Cells were then fixed with 4%PFA in PBS at 24 and 48 h post-infection, and washed several times in PBS before proceeding for Raman confocal microscopy.

A similar procedure was followed for the infection of Vero E6 cells with MeV virus (Schwartz strain), in six-well plates using an MOI of 0,1, as described previously[56]. At 90 min post-infection, the virus was aspirated, and a fresh medium was added. MeV-infected Vero E6 cells were then fixed at 24 h post-infection with 4% PFA in PBS and washed several times in PBS before proceeding for Raman confocal microscopy. All the infection assays were performed in triplicate conditions.

**Immunoblots**. Vero E6 cells were infected with SARS-CoV-2 (MOI = 0.1) for, 24–48 h. Cells were washed twice in Saline Phosphate Buffer (PBS), detached, pelleted, and lysed in RIPA buffer for western blot analysis. Total protein concentration was calculated using a Bradford protein assay kit (Thermo Fisher). Twenty micrograms of total cell lysates were diluted in Laemmli buffer and proteins were separated by SDS-PAGE on 8% acrylamide gels. Gels were transferred to PVDF membrane (Amersham) using wet transfer with Tris-glycine-methanol buffer. Membranes were then washed in TBS, blocked with 5% milk in TBS-Tween 0.1%, and incubated with primary antibodies against the spike S protein (Gentex), the N-protein (Gentex), or the hACE2-protein (Gentex). After washing, the membranes were incubated with HRP conjugated anti-mouse antibodies for 1 h at room temperature (RT), then washed, incubated with ECL reagent (Amersham), and imaged using a Chemidoc Imager (Biorad).

**Immunofluorescence confocal microscopy**. Vero E6 cells seeded on glass cover-slips were infected with SARS-CoV-2 at MOI = 0.1. After 24 and 48 h post-infection, cells were washed with PBS and fixed in 4% paraformaldehyde in PBS for 15 min at room temperature, followed by permeabilization with 0.2% Triton X-100 in PBS for 5 min and blocking in 2% BSA in PBS for 15 min. Incubation with primary antibodies anti-SARS-CoV-2 rabbit membrane (M) protein (Tebu, cat# 039100-401-A55) were performed for 1 h at RT. After washing with PBS, cells were incubated with secondary antibodies AF568-labeled goat-anti-rabbit and AF488-labeled Phalloidin for 1 h at RT, then mounted in media prolong gold antifade reagent (Thermo Fisher) for confocal microscopy. Confocal fluorescence images were generated using an LSM800 confocal laser-scanning microscope (Zeiss) equipped with a 63X, 1.4 NA oil objective. All the images were processed with ImageJ/Fiji software.

**Electron microscopy**. Vero E6 cells were infected with $1.10^6$ PFU of SARS-CoV-2 for 24 h. Cells were fixed with 2,5% (v/v) glutaraldehyde in PHEM buffer and postfixed in osmium tetroxide 1% / $K_4Fe(CN)_6$ 0,8%, at room temperature for 1 h for each treatment[55]. The samples were then dehydrated in successive ethanol baths (50/70/90/100%) and infiltrated with propylene oxide/ EMbed812 mixes before embedding. Seventy-nanometer ultrathin cuts were made on a PTXL ultramicrotome (RMC, France), stained with OTE/lead citrate, and observed on a Tecnai G2 F20 (200 kV, FEG) TEM at the Electron Microscopy Facility MRI-COMET, INM, Plate-Forme Montpellier RIO Imaging, Biocampus, Montpellier.

**Raman spectroscopy imaging**. WITec Confocal Raman Microscope Alpha 300 R (WITec Inc., Ulm, Germany) is used to collect Raman spectra. a frequency-doubled Nd:YAG laser (Newport, Evry, France) with 532 nm wavelength and 50 mW power provided sample excitation. A 60x NIKON water immersion objective (numerical aperture of NA = 1.0.) focused laser beam on PBS immerged cells. an electron-multiplying charge-coupled device (EMCCD) camera (DU 970 N-BV353, Andor, Hartford, USA) captured the scattered signals. Using the formula $r_{lateral} = 1.22 \cdot \lambda_{laser}/2 \cdot NA$ gives the spatial resolution of the system 325 nm. For the axial resolution, $r_{axial} = 1.4 \cdot \lambda_{laser} \cdot n/NA^2$ (where n is the index of refraction 1.33 for the water-based objective) gives 991 nm. WITec Image Plus software performed data acquisition and processing. Calcium fluoride ($CaF_2$) substrate was employed due to its characteristic Raman peak at 320 $cm^{-1}$ to avoid extra Raman signal interfering with cells signature. Each Raman scan contains more than twenty thousand single spectra. The recorded Raman spectra collected from each voxel (300 nm × 300 nm × 900 nm) contain the sample biochemical fingerprint under the laser spot of 1 μm.

### Raman spectral data analysis

*Reconstructed Raman images using KMCA*. To generate a false-color Raman image, advanced data analysis based on two steps was applied. The First method is collecting the integrated Raman intensities of the C-H stretching mode to visualize the cell image. The second method k-mean clusters analysis (KMCA) contains the post-processing of the recorded spectra. KMCA is a simple algorithm to divide all collected spectra into k mutual groups. It considers each data set—Raman spectrum—as a spot in multidimensional space. KMCA splits these spots in groups within each, spots—Raman spectra—are as close to each other as possible, and as far from spots in other groups as possible. Based on the presence of specific peaks or their relative intensities, the k clusters are used to identify Raman information or separation of certain cellular organelles (including the nucleus, nucleolus, mitochondria, Golgi, and cytoplasm). KMCA is done using WITec Project Plus software (Ulm, Germany).

**Raman spectra preprocessing**. Raman spectra were extracted from the different intracellular components, namely nucleus, Golgi-mitochondria bodies, and cytoplasm. The infection experiments were performed in triplicate and from each experimental batch, 40 spectra/conditions were extracted from each intracellular component. The Raman spectra were preprocessed using GNU R platform[57] with an in-house built script. Before preprocessing, Raman spectra were truncated from 360 to 3050 $cm^{-1}$, and the background was subtracted using the Sensitive Nonlinear Iterative Peak (SNIP)[58,59] algorithm. The background-subtracted spectra were further truncated and spectral range from 400 to 1800 $cm^{-1}$ and 2700 to 3050 $cm^{-1}$ and the spectra were vector normalized. The Raman difference spectra were generated and plotted using OriginPro v2020 software (OriginLab Corporation, Northampton, MA, USA).

**Principal component analysis (PCA)**. The PCA was performed on the normalized Raman spectra and principal component (PC) score plots and loadings coefficients were generated. Raman spectra along with PCA score plots and loadings were plotted using Origin software.

**Support vector machine (SVM)**. The support vector machine (SVM) algorithm was used to build the Raman models. For the SVM analysis, Raman spectral data from three biological replication experiments were used. Firstly, PCA analysis was performed, separately, on the Raman spectra extracted from the nucleus, Golgi-mitochondria, and cytoplasm. The first 13 principal components (PCs) obtained from PCA analysis were used as an input for the SVM analysis. Three separate, two-class SVM models were built to differentiate between (1) noninfected and MeV-infected Vero E6 cells, (2) noninfected and SARS-CoV-2-infected Vero E6 cells, and (3) MeV and SARS-CoV-2-infected Vero E6 cells. The parameters chosen for SVM analysis are as follow: gamma = 1e-4, cost = 200, kernel = "radial", type = "C-classification". The SVM analysis was cross-validated using tenfold cross-validation. The SVM decision value plots were generated using Origin software.

**Reporting summary**. Further information on research design is available in the Nature Research Reporting Summary linked to this article.

## Data availability
The datasets generated during and/or analysed during the current study are available from the corresponding author on reasonable request.

## Code availability
The GNU R code used during the study for the analysis of the Raman spectroscopy data were available from the First author on reasonable request or can be accessed via RamanMetrix software Leibniz-IPHT 2020 (https://ramanmetrix.eu).

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

## Acknowledgements
Raman microscopy analysis during this study was realized using the EDMOS platform which was created with the financial support of the Region Occitanie (France) and the European Regional Development Fund (ERDF). We thank Pasteur Institute for providing us with the SARS-CoV-2 strain and the CEMIPAI for the SARS-CoV-2 laboratory facility. This study was supported by the CNRS and Montpellier University through a Montpellier Université d'Excellence (MUSE) support and by ANR-21-CO15-0007 and ANR-20-COVI-000 and Fondation de France to B.H. Funding from the Deutsche Forschungsgemeinschaft (DFG, German Research Foundation) under Germany's Excellence Strategy—EXC 2051—Project-ID 390713860 is acknowledged.

## Author contributions
H.S. performed Raman measurements and KMCA data analysis, wrote the manuscript and A.R. performed the PCA, SVM data analysis, and contributed to draft preparation. P.M., D.M., and S.M. performed viral stock, cell culture, infection, and immunoblots; D.M. performed immunofluorescence confocal microscopy; A.N. performed TEM sample preparation and imaging; B.H. and D.M. edited the manuscript and raised funding. J.P. and F.C. planned the study and reviewed the manuscript.

## Competing interests
The authors declare no competing interests.
