## [Peer review file · Communications Chemistry]

Reviewers' comments:

Reviewer #1 (Remarks to the Author):

The authors study cellular response of SARS-CoV-2 infection into cultured Vero E6 cells with Raman imaging microscopy. The results in the present study demonstrate the potential of Raman analysis in the virus infection study and are quite valuable. This reviewer suggests a several improvements to clarify the manuscript before the publication.

The value of MOI seems too small. The cell which can infected with the virus is less than 10 %, probably 2~3%. Besides, a part of the viruses sometimes takes a dormant state after the infection. How did the authors find the infected cells for Raman measurements? The authors should confirm the rate of virus infection with a conventional method, such as immune-staining.

Or, do the authors mean that the multi-generations of infections have already takes place within only 24 h? In this case, there are cells at varied states of the virus infection. The authors must explain how and which state of the cells they select to analyze.

The authors must clarify how many cells have been measured in each batch. They also should explain what "background" means. Was the background spectrum obtained at every dish? Does it come only from the aqueous solution? Was there background originated from fluorescence? They are necessary to ensure the result of multivariate analysis.

According to Fig.3 and Fig.S2, the reproducibility of the present experiments is low. Does it come from difficulty of controlling the sample condition, or come from any instrumental reason?

Reviewer #2 (Remarks to the Author):

Specific intracellular signature of SARS-CoV-2 infection using confocal Raman microscopy

The manuscript titled "Specific intracellular signature of SARS-CoV-2 infection using confocal Raman microscopy" describes biochemical modification due to SARS-CoV-2 infection in cells by using confocal Raman spectroscopy. Biomolecular modification of cellular organelles after SARS-CoV-2 infections has been compared to another RNA virus, the measles virus (MeV) to analyze virus-specific intracellular modifications. The list of obtained biochemical changes is provided to help in better understanding the cellular signature of virus infection.

Below, there are some comments and suggestions that have been provided for the authors to enhance the clarity of the manuscript:

1. Authors demonstrated the biochemical modifications that occurred during virus infection with both Raman mapping images and spectra obtained from subcellular organelles. In studies investigating virus-infected cells, only spectra from cells are available, Raman mapping images are not available. The presence of Raman mapping images and the Raman spectra fills the gap in the literature to understand virus-host interactions, develop novel diagnostic markers, and observe the changes that occur when the virus infects the cells. Especially in pandemics where new diagnostic methods will be needed, investigating infected cells for early detection of SARS-CoV-2 is a novelty and will be a guide for further studies. However, there is a wonder in why the authors only used images 24 hours after infection with the virus? What changes might occur after 6h, 12h, or 48h?

2. Authors stated that the differences between infected and non-infected cells in the cytoplasm, Golgi-mitochondria bodies, and nucleus can be observed mainly in the Raman peak profile of the C-H stretching region (2800-3050 cm^{-1}). Is the cause of these results due to using a 532 nm wavelength laser? Does using the 785 nm laser might change the results?

3. Authors should discuss the following SARS-CoV-2 and other virus detection with Raman

spectroscopy-related articles to the manuscript.

- Akdeniz, M., Ciloglu, F. U., Tunc, C. U., Yilmaz, U., Kanarya, D., Atalay, P., & Aydin, O. (2022). Investigation of mammalian cells expressing SARS-CoV-2 proteins by surface-enhanced Raman scattering and multivariate analysis. *Analyst*, 147(6), 1213-1221.
- Desai, S., Mishra, S. V., Joshi, A., Sarkar, D., Hole, A., Mishra, R., ... & Dutt, A. (2020). Raman spectroscopy-based detection of RNA viruses in saliva: A preliminary report. *Journal of Biophotonics*, 13(10), e202000189.

4. Using unsupervised machine learning to distinguish between non-infected cells and infected cells (MeV and SARS-CoV-2) in different subcellular components (nucleus, mitochondria, and cytoplasm) has demonstrated good classification results. The exploit of PCA to minimize the high-dimensionality of the Raman spectra and then the use of SVM to classify the reduced spectra have provided significant evidence for the ability of machine learning not just to study a single cell but also to study its subcomponent. The results of the classification are pretty good since the accuracies after using 10-fold cross-validation are 99%, 97%, and 98 for binary classifier of non-infected and MeV-infected cells; non-infected and SARS-CoV-2 infected cells; and SARS-CoV-2 MeV-infected cells, respectively. Although there are some interferences when the three classes were classified together, the accuracy of this classifier is good as well. Could the authors give a brief explanation about the reduction in the accuracy of the three-class classifier?

5. To check this idea works for most cell lines other than only Vero E6 cell lines, other cell lines like virus-infected HEK 293 or other ones should be included in the experimental part. This will give us to see whether or not this analysis will work in all cell lines differently or the Raman signal will be the same and the analysis might be suspected in the discrimination of cellular compartments. So, the same procedures should apply on COVID-19 infected different cell lines.

6. There are some minor typing and punctuation errors that should be addressed.

Reviewer #3 (Remarks to the Author):

The authors present a Raman spectroscopy approach to study virus infection on cells. The approach and results are interesting. Here are some comments.

1. What's the substrate for SERS? In the manuscript, it says it's a SERS but I can't find what's the SERS substrate or it just a Raman spectroscopy?
2. How repeatable of spectrums for different compartments of cells? In figure 2. there are one spectrum but methods say thousand of the spectrum were collected. Are those spectrums overlap to each other? How repeatable of spectrum for each compartment?
3. For table 1, how these peak assignments are achieved? Is there any reference?

Dear Editor of Nature Communication Chemistry

On behalf of all authors, I would like to thank you and the reviewers to give us the opportunity to get examined our research results. We appreciate the reviewers' time and efforts to upgrade the manuscript. We have provided reply to the reviewers' comments and questions, point by point, and hopefully, our modified manuscript and answers meet their satisfaction.

Reviewer #1 (Remarks to the Author):

The authors study cellular response of SARS-COV-2 infection into cultured Vero E6 cells with Raman imaging microscopy. The results in the present study demonstrate the potential of Raman analysis in the virus infection study and are quite valuable. This reviewer suggests a several improvements to clarify the manuscript before the publication.

Ans: We thank the Reviewer for the positive evaluation of our manuscript. We have addressed different remarks point-by-point.

The value of MOI seems too small. The cell which can be infected with the virus is less than 10 %, probably 2~3%. Besides, a part of the viruses sometimes takes a dormant state after the infection. How did the authors find the infected cells for Raman measurements? The authors should confirm the rate of virus infection with a conventional method, such as immune-staining.

Ans:

1- HEK293 cell line is not a good model cell line for SARS-CoV-2 infection as they don't express the virus receptor ACE2. VeroE6 cells have natural ACE2 receptor expression. See Supplemental Figure S1 A

2- the chosen value for MOI was 0.1 and is not too small to infect VeroE6 cells with the SARS-CoV-2 Wuhan strain. The infection and propagation of SARS-CoV-2 in vitro is very rapid and at 72h, for MOI=0.1, almost all the infected cells were dead and destroyed. SARS-CoV-2 is a lytic virus: it is how we evaluate drug protection by measuring cell viability after infection, with the Cytopathic Effect (CPE) of the virus. Thus at 24h, since almost all the VeroE6 cells are infected, it was easy to find them. At 24h or 48h, MOI=0.1 was the good timing to do Raman confocal analysis, before cells were destroyed by infection. In addition, we thus present Immunofluorescence staining of SARS-Cov2 at 24h and 48h post-infection (MOI=0.1) in VeroE6 cells showing all infected cells. See new Supplemental Figure S1 in the revised manuscript (supplementary data).

SARS-CoV-2 Infection in VeroE6 cells

Supplemental Figure S1: Characterization of SARS-CoV-2 infected VeroE6 cells.

- (A) Cellular human ACE2 and viral gene expression analysis of SARS-CoV-2 infected VeroE6 cells using immunoblots. Western-blot analysis for human ACE2 expression in HEK293T (lane 1) and in VeroE6 (lane 2) cell lysates, and in infected (lane 4) versus non-infected (lane 3) VeroE6 cells. Western blots of the SARS-CoV-2 Spike and Nucleocapside N proteins in control (lane 3) and infected (lane 4) VeroE6 cells 48h post-infection with SARS-CoV-2 (Wuhan) at MOI= 0.1.
- (B) Immuno-fluorescence images of infected VeroE6 cells 24h or 48h post-infection with SARS-CoV-2 (Wuhan) using confocal fluorescence microscopy. For imaging the virus, rabbit anti-M primary antibody and then secondary antibody anti-rabbit Alexa568 (red) were used. For imaging the cells, F-actin labelling with Phalloidin-Alexa 488 (green) was used on fixed cells. The merged image is shown revealing all infected cells, in conditions similar to the confocal Raman spectroscopy experiment.

The modified manuscript in red (Results part) , is added to make it clear. Relevant material method part is added.

The SARS-CoV-2 infection of VeroE6 cells was verified by immunoblots, immunofluorescence, and electron microscopy (supplement Figure S1). Immunoblots anti-ACE2 show the presence of the virus receptor expressed in VeroE6 cells, compatible with SARS-CoV-2 infection, before and after viral infection, in contrary to HEK293T cells that do not harbor ACE2 receptor. Anti-Spike and anti-N antibodies reveal the abundant expression of the CoV-2 Spike and Nucleocapsid N proteins in cell lysates of infected cells at MOI=0.1 (supplement Figure S1A). In parallel, the sample of SARS-CoV-2 infected VeroE6 cells were analyzed by immunofluorescence at 24h and 48h post-infection showing that the majority, if not all, of the cells were infected by SARS-CoV-2 (supplement Figure S1B). By electron microscopy, we also follow the virus infection and observe the visible intracellular modifications due to the virus, 24h post-infection (pi) (supplement Figure S1C). Transmission electron microscopy (TEM) images showed viral particles at the cell plasma membrane signing viral production (supplement Figure S1C-a) and/or further virus attachment for cell entry. In addition, intracellular organelles filled with viral factories were recognizable inside the cytoplasm, near the nucleus and mitochondria, most probably reflecting the double-membrane vesicles or spherules replicative factories of SARS-CoV-2 derived from the ER-Golgi apparatus (supplement Figure S1C-b) as previously observed and described^{36,37}. These TEM images confirmed that virus particles were produced by the infected cells and that they can be visualized at the cell plasma membrane and 24h post-infection.

Or, do the authors mean that the multi-generations of infections have already takes place within only 24 h? In this case, there are cells at varied states of the virus infection. The authors must explain how and which state of the cells they select to analyze.

Ans: Yes, multiple rounds of infection have already taken place within 24h post-infection at MOI=0.1. The selected cells were infected, as revealed by immune-fluorescence staining. See the supplement Figure S1 which has been added to the revised manuscript.

The authors must clarify how many cells have been measured in each batch. They also should explain what “background” means. Was the background spectrum obtained at every dish? Does it come only from the aqueous solution? Was there background originated from fluorescence? They are necessary to ensure the result of multivariate analysis.

Ans: In each batch between 20-30 cells were scanned (each scan contains at least twenty-thousand spectra) within infected and control groups. The spectral background is the contributions coming from various factors including fluorescence. The other contributing factors are noise from the instrument (shot noise, dark current), and environmental noise such as room light contributions. However, to account for these variabilities the spectra are pre-processed before applying the multivariate analysis, where the SNIP algorithm is applied for background

subtraction followed by vector normalization. These pre-processing steps allow minimizing/removing the external factors influencing the Raman spectra and make them comparable ensuring unbiased results from the multivariate analysis.

According to Fig.3 and Fig.S2, the reproducibility of the present experiments is low. Does it come from difficulty of controlling the sample condition, or come from any instrumental reason?

Ans: The principal component analysis (PCA) analysis takes into account total variations which accompany a measured Raman spectrum such as the variation in the sample condition (cell culture, infection assay, etc.) and instrumental variations (temperature variability, instrument alignment, fluctuation in laser power, variation in instrumental noise, etc.). Usually, the PCA method is not powerful enough to process data from multiple batches if the batch to batch variation is higher than the differences expected within the sample groups. In such a scenario output from PCA is taken and a supervised multivariate analysis method is applied such as a support vector machine (SVM). We used PCA as an unsupervised method to show that for each of the individual experiments we can differentiate between infected and non-infected and also can detect the type of virus causing the infection. To show the reproducibility of our results, we applied the SVM method where all the experimental replicates (multiple batches) were analyzed in a combined fashion. The combined analysis also yielded high accuracy for the differentiation of infected and non-infected cells.

Reviewer #2 (Remarks to the Author):

Specific intracellular signature of SARS-CoV-2 infection using confocal Raman microscopy
The manuscript titled "Specific intracellular signature of SARS-CoV-2 infection using confocal Raman microscopy" describes biochemical modification due to SARS-CoV-2 infection in cells by using confocal Raman spectroscopy. Biomolecular modification of cellular organelles after SARS-CoV-2 infections has been compared to another RNA virus, the measles virus (MeV) to analyze virus-specific intracellular modifications. The list of obtained biochemical changes is provided to help in better understanding the cellular signature of virus infection.

Below, there are some comments and suggestions that have been provided for the authors to enhance the clarity of the manuscript:

We thank the Reviewer for the positive evaluation of our manuscript. We have addressed different remarks point-by-point.

1. Authors demonstrated the biochemical modifications that occurred during virus infection with both Raman mapping images and spectra obtained from subcellular organelles. In studies investigating virus-infected cells, only spectra from cells are available, Raman mapping images are not available. The presence of Raman mapping images and the Raman spectra fills the gap in the literature to understand virus-host interactions, develop novel diagnostic markers, and observe the changes that occur when the virus infects the cells. Especially in pandemics where new diagnostic methods will be needed, investigating infected cells for early detection of SARS-

CoV-2 is a novelty and will be a guide for further studies. However, there is a wonder in why the authors only used images 24 hours after infection with the virus? What changes might occur after 6h, 12h, or 48h?

Ans: We have indeed performed the Raman mapping images at 48h post-infection as well and they are provided in Figures R1 and R2 below. We like to emphasize that each Raman image is reconstructed using more than twenty-two thousand spectra, therefore all the intracellular information being used were extracted from the Raman spectra. In Figure 1, Raman images of SARS-CoV-2 and MeV-infected (24h post-infection) and non-infected Vero E6 cells are shown. For reconstruction of these images the spectral region $2800-3000\text{ cm}^{-1}$ (which belongs to lipids and proteins) is selected. The yellow color corresponds to the highest intensities of lipids and proteins and dark shades for the lowest intensities as indicated by the color scale bar in these false-color images.

The infection of the cells at 6h post-infection was not performed (mainly because at early infection, it would have been very difficult to discriminate between infected and non-infected cells). The experimental conditions used for the Raman spectroscopy, 24h or 48h pi, we were sure that all cells were infected (see supplement Figure S1).

The viral infection after 48h was investigated and we observed similar results as after 24h post-infection because all the cells were infected (see supplemental Figures) and the intracellular changes occurring due to infection had reached maturity at 48h pi. In the below Figure R1, we have presented the PCA analysis of the Vero E6 cells 48h post-infection. As viral infection at 48h leads to the important cytopathic effect, which may interfere with the Raman spectrum, we focused only on the 24h pi in the manuscript.

Figure R1: Raman confocal images of SARS-CoV-2 (48h pi) Vero E6 cells. In upper row images, the light yellow color corresponds to the highest intensities of lipids and proteins and dark shades for the lowest intensities of C-H signals (see the scale bar on right). The lower row present the KMCA images, where the

nucleolus are presented in dark green, the nucleus in turquoise, mitochondria, and Golgi are in blue, cytoplasm in pink. Scale bars are variable between 4-10 μm as indicated.

The comparability of the results obtained for 24h and 48h pi can be observed from Figures R1 and R2, However, as mentioned above it can be seen that the cells start to undergo cytoplasmic damage. Hence, 24h infection is been chosen to present as the optimized infection time point to observe the viral infection-related changes within the VeroE6 cells.

Figure R2: Mean Raman spectra from the cytoplasm, Golgi-mitochondria bodies, and nucleus region shown for non-infected (Ni 48h) and 48h post SARS-CoV-2 infected Vero E6 cells (i 48h) along with the standard deviation. A) Raman spectra extracted from cytoplasm region, B) Raman spectra extracted from Golgi-mitochondria bodies, and C) Raman spectra extracted from nucleus region. D to F) Score plot obtained using principal component analysis (PCA) to differentiate between non-infected and 48h post SARS-CoV-2 infected cells (i 48h) for D) cytoplasm, E) Golgi-mitochondria and F) nucleus region of the Vero E6 cells. G to I) Principal component (PC) loadings obtained after PCA analysis of non-infected and SARS-CoV-2 infected cells G) cytoplasm, H) Golgi-mitochondria and I) nucleus region of the Vero E6 cells. The Raman spectra and the PC loadings are shifted on the y-axis for sake of clarity.

2. Authors stated that the differences between infected and non-infected cells in the cytoplasm, Golgi-mitochondria bodies, and nucleus can be observed mainly in the Raman peak profile of the C-H stretching region (2800-3050 cm⁻¹). Is the cause of these results due to using a 532 nm wavelength laser? Does using the 785 nm laser might change the results?

Ans: The changes observed in the CH stretching region are independent of the excitation laser used. By using a 785nm excitation wavelength one expects similar results, however, the detector used for the 785nm laser has lower sensitivity in the CH stretching region resulting in lower Raman peak intensity for the CH stretching vibrations.

3. Authors should discuss the following SARS-CoV-2 and other virus detection with Raman spectroscopy-related articles to the manuscript.

• Akdeniz, M., Ciloglu, F. U., Tunc, C. U., Yilmaz, U., Kanarya, D., Atalay, P., & Aydin, O. (2022). Investigation of mammalian cells expressing SARS-CoV-2 proteins by surface-enhanced Raman scattering and multivariate analysis. *Analyst*, 147(6), 1213-1221.

• Desai, S., Mishra, S. V., Joshi, A., Sarkar, D., Hole, A., Mishra, R., ... & Dutt, A. (2020). Raman spectroscopy-based detection of RNA viruses in saliva: A preliminary report. *Journal of Biophotonics*, 13(10), e202000189.

Ans: We thank the Reviewer for the suggestions, we have included the additional literature in the revised manuscript. the modified/added text in red is included in the manuscript.

Different viral dilutions to mimic saliva conditions were been used to detect RNA virus using Raman spectroscopy²⁷ (Desai et al). Recent studies reported the detection of SARS-CoV-2 using Surface Enhanced Raman Scattering (SERS)^{1,28-33}. SERS results coupled with PCA analysis is been introduced as a powerful tool to detect the virus³⁴(Akdeniz et al). In this study gold nanoparticles are been used as a substrate to create the plasmons for surface enhanced Raman signals on transfected HEK293 cells.

4. Using unsupervised machine learning to distinguish between non-infected cells and infected cells(MeV and SARS-CoV-2) in different subcells components (nucleus, mitochondria, and cytoplasm) has demonstrated good classification results. The exploit of PCA to minimize the high-dimensionality of the Raman spectra and then the use of SVM to classify the reduced spectra have provided significant evidence for the ability of machine learning not just to study a single cell but also to study its subcomponent. The results of the classification are pretty good since the accuracies after using 10-fold cross-validation are 99%, 97%, and 98 for binary classifier of non-infected and MeV-infected cells; non-infected and SARS-CoV-2 infected cells; and SARS-CoV-2 MeV-infected cells, respectively. Although there are some interferences when the three classes were classified together, the accuracy of this classifier is good as well. Could the authors give a brief explanation about the reduction in the accuracy of the three-class classifier?

Ans: In the three-class, we are distinguishing also the type of virus causing the infection hence the discrimination power is higher than in the two-class model. A misidentification of virus-type infection will not influence the outcome of the two-class model just of the three-class model. Therefore a small decrease in the classification accuracy is expected when going from a two- to three-class model.

5. To check this idea works for most cell lines other than only Vero E6 cell lines, other cell lines like virus-infected HEK 293 or other ones should be included in the experimental part. This will give us to see whether or not this analysis will work in all cell lines differently or the Raman signal will be the same and the analysis might be suspected in the discrimination of cellular compartments. So, the same procedures should apply on COVID-19 infected different cell lines.

Ans: We agree with the reviewer that using an additional cell line could be beneficial for the confirmation of obtained results. However, HEK293 cells do not express SARS-CoV-2 receptor ACE2 and are thus not susceptible to the infection. Nevertheless, the other group, mentioned by this reviewer, has very recently performed the analysis of the effect of transfected SARS-CoV-2 proteins in HEK 293 cells, using Surface-enhanced Raman scattering (Akdeniz et al, PMID: **35212693**). The results obtained in their study are in agreement with our study. This is now additionally discussed in the manuscript.

SERS results coupled with PCA analysis is been introduced as a powerful tool to detect the virus³⁴(Akdeniz et al). In this study gold nanoparticles are been used as a substrate to create the plasmons for surface enhanced Raman signals on transfected HEK293 cells.

6. There are some minor typing and punctuation errors that should be addressed.

Ans: We thank the Reviewer, the final version of the manuscript is been revised and proofread to avoid typo errors.

Reviewer #3 (Remarks to the Author):

The authors present a Raman spectroscopy approach to study virus infection on cells. The approach and results are interesting. Here are some comments.

We thank the Reviewer for the positive evaluation of our manuscript. We have addressed different remarks point-by-point.

1. What's the substrate for SERS? In the manuscript, it says it's a SERS but I can't find what's the SERS substrate or it just a Raman spectroscopy?

Ans: In the current manuscript Raman spectroscopy study has been shown. We apologize for any confusion.

2. How repeatable of spectrums for different compartments of cells? In figure 2. there are one spectrum but methods say thousand of the spectrum were collected. Are those spectrums overlap to each other? How repeatable of spectrum for each compartment?

Ans:

Every cell (<30 μ m in diameter) was scanned with a scan zone area of 150*150 or 200*200 voxels (300nm*300nm*900nm X-Y-Z respectively). The laser excites all molecules within this volume and a single spectrum from this volume is collected which contains all chemical bonds from this zone. All the scan zone from a given cell is used and a false-color Raman image is generated. Each reconstructed Raman cell image contains at least twenty-two thousand spectra.

Further for the analysis, guided by the false-color Raman cell image generated using KMCA, averaged Raman spectra are extracted from the nucleus, mitochondria-Golgi and cytoplasm regions. The regions chosen to extract the Raman spectrum contain multiple scan zone, hence each extracted spectrum is an average of multiple Raman spectra (<10 single spectra).

3. For table 1, how these peak assignments are achieved? Is there any reference?

Ans: The Raman peak assignments are based on the literature study. For clarity, we have included in the Table heading the literature references.

REVIEWERS' COMMENTS:

Reviewer #1 (Remarks to the Author):

The manuscript has been improved well. The answers to the reviewer are enough informative to understand their study. This reviewer suggest to publish the manuscript.

Reviewer #2 (Remarks to the Author):

Specific intracellular signature of SARS-CoV-2 infection using confocal Raman microscopy
The manuscript titled 'Specific intracellular signature of SARS-CoV-2 infection using confocal Raman microscopy' describes biochemical modification due to SARS-CoV-2 infection in cells by using confocal Raman spectroscopy. Biomolecular modification of cellular organelles after SARS-CoV-2 infections has been compared to another RNA virus, the measles virus (MeV), to analyze virus-specific intracellular modifications. The list of obtained biochemical changes is provided to help in better understanding the cellular signature of virus infection.

- In the previous report, the manuscript was reviewed and some comments such as the effect of different incubation times for virus infection, the effect of using different lasers, and the accuracy of the three-class classifier. The given responses for these comments and the revised manuscript have been reviewed. It has been observed that the responses and additions to the manuscript were sufficient. Generally, revisions have been made in the line with the comments we have made. The manuscript can be accepted.

Reviewer #3 (Remarks to the Author):

All comments are addressed well. I have no comments.